

# Crescent-shaped electron velocity distribution functions formed at the edges of plasma jets interacting with a tangential discontinuity

Gabriel Voitcu[1], Marius Echim[1,2]

[1]Institute of Space Science, Magurele, 077125, Romania
[2]Belgian Institute of Space Aeronomy, Brussels, B-1180, Belgium

*Correspondence to*: Gabriel Voitcu (gabi@spacescience.ro)

**Abstract.** In this paper we discuss numerical simulations that illustrate a physical mechanism leading to the formation of crescent-shaped electron velocity distribution functions at the edges of a high-speed plasma jet impacting on a thin, steep and impenetrable tangential discontinuity with no magnetic shear. We use three-dimensional particle-in-cell simulations to
compute the velocity distribution function of electrons in different areas of the plasma jet and at different phases of the interaction with the discontinuity. The simulation setup corresponds to an idealised, yet relevant, magnetic configuration likely to be observed at the frontside magnetopause under northward interplanetary magnetic field. The combined effect of the gradient-B drift and the remote sensing of large Larmor radius electrons leads to the formation of crescent-shaped electron velocity distribution functions. We provide examples of such distributions "measured" by a virtual satellite
launched into the simulation domain.

## 1 Introduction

The terrestrial magnetosheath is often populated by high-speed plasma jets (also called plasmoids or plasma irregularities) streaming towards the frontside magnetopause (e.g., Dmitriev and Suvorova, 2012, 2015; Karlsson et al., 2012; Gunell et al., 2014) and impact it several times per hour (Plaschke et al., 2016). These finite-size plasma structures
have an excess of antisunward dynamic pressure with respect to the ambient magnetosheath (e.g., Hietala et al., 2012; Savin et al., 2012; Archer and Horbury, 2013; Plaschke et al., 2013, 2017; Karlsson et al., 2018) and their interaction with the magnetopause can trigger local and global magnetospheric and geophysical effects (e.g., Plaschke et al., 2009; Hietala et al., 2012, 2018). Dmitriev and Suvorova (2015) shows that >60% of the jets detected by the Time History of Events and Macroscale Interactions during Substorms (THEMIS) mission do penetrate inside the Earth's magnetosphere, while the rest
of <40% are not able to cross over the magnetopause. Some of the penetrating plasmoids were intersected in-situ by spacecraft deep inside the magnetosphere (e.g., Lundin and Aparicio, 1982; Woch and Lundin, 1991, 1992; Yamauchi et al., 1993; Lu et al., 2004; Gunell et al., 2012; Shi et al., 2013; Lyatsky et al., 2016).

The transport of plasma jets/clouds/blobs across various magnetic field configurations has been investigated with magnetohydrodynamic (MHD) (e.g., Ma et al., 1991; Dai and Woodward, 1994, 1995, 1998; Huba, 1996), hybrid (e.g.,
Savoini et al., 1994; Palmroth et al., 2018) and kinetic (e.g., Galvez, 1987; Galvez et al., 1988; Livesey and Pritchett, 1989; Galvez and Borovsky, 1991; Cai and Buneman, 1992; Neubert et al., 1992; Hurtig et al., 2003; Gunell et al., 2009) numerical simulations (see also the review by Echim and Lemaire, 2000). These previous simulations did not reveal all the fundamental aspects of jets dynamics due to the inherent physical and/or geometrical limitations. For instance, the MHD and hybrid simulations do not quantify the effect of self-polarization (Schmidt, 1960) which is purely kinetic and plays a crucial
role for plasma motion across transverse magnetic fields. On the other hand, the particle-in-cell (PIC) simulations were limited by the existing computing resources and none of them succeeded to evaluate the interaction of a three-dimensional plasma jet with a tangential discontinuity (TD). Recently, Voitcu and Echim (2016, 2017) used a 3-D PIC code to simulate the dynamics of high-speed plasma jets interacting with magnetic fields typical for a tangential discontinuity with no shear. Such a configuration is similar to the magnetopause under a northward interplanetary magnetic field (IMF). It has been

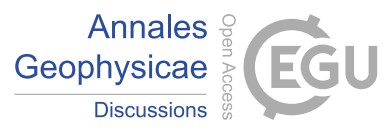

shown that the plasma-field interaction process is controlled by the dynamic and kinetic pressure of the high-speed jet, its electric polarizability and the magnetic jump at the TD. It was also shown that when the dynamic pressure is large enough the jet can cross the TD, as implied by kinetic models like the impulsive penetration mechanism (Lemaire, 1977, 1985; see also Lemaire and Roth, 1991). When the dynamic pressure is small, the jet is stopped by the TD, consistent with THEMIS

observations of jets reported by Dmitriev and Suvorova (2015). The dynamics of finite-size plasma jets/clouds across transverse magnetic fields has been described also with Vlasov equilibrium solutions by Echim and Lemaire (2005).

        In this paper we discuss the kinetic structure of a low-beta plasma jet impacting on a thin, steep and impenetrable tangential discontinuity with no magnetic shear. The analysis is based on 3-D electromagnetic PIC simulations. We consider here an idealised, yet relevant, simulation setup that corresponds to a magnetosheath plasma jet streaming towards the

frontside region of the Earth's magnetopause and interacting with it under a northward interplanetary magnetic field. The plasma jet has a relatively small momentum and it cannot move across the rather steep discontinuity; it is repelled and split in two plasma streams moving in opposite directions, tangentially to the TD surface, as discussed in detail in a previous publication (Voitcu and Echim, 2017). However, kinetic processes observed during the interaction between the jet and the discontinuity lead to the formation of crescent-shaped electron velocity distribution functions (VDFs).

Crescent-like VDFs were observed for ions in the Earth's magnetotail, close to the neutral sheet and plasma sheet boundary layer (e.g., Nakamura et al., 1991, 1992; Wilber et al., 2004), as well as in the magnetosheath, close to the magnetopause (e.g., Marcucci et al., 2004). The crescent-shaped ion VDFs have been associated with various finite Larmor radius effects (e.g., Lee et al., 2004; Marcucci et al., 2004; Voitcu and Echim, 2012; Voitcu et al., 2012). Recently, crescent-shaped VDFs were observed by the Magnetospheric Multiscale (MMS) mission for electrons close to the dayside

magnetopause (Burch et al., 2016) and were associated to meandering electron orbits near magnetic field reversal in the reconnection sites (Bessho et al., 2016; Lapenta et al., 2017). Egedal et al. (2016) proposed a two-dimensional theoretical model for the formation of crescent-shaped electron VDFs in asymmetric magnetic reconnection. The magnetic configuration used in our study does not involve the presence of magnetic field reversal regions, as in, for instance, Bessho et al. (2016), Egedal et al. (2016) or Lapenta et al. (2017), since we consider a tangential discontinuity with no shear across

it. The physical mechanism revealed by our PIC simulations leads to crescent-shaped VDFs and is specific to the interaction between a plasma jet and a parallel non-uniform magnetic field; it does not require magnetic reconnection. This paper provides new insight on the kinetic structure of high-speed plasma jets moving in the vicinity of unidirectional magnetic barriers. We emphasize the role of finite Larmor radius effects that give rise to crescent-shaped electron distributions in the space of perpendicular velocities. Such kind of electron VDFs can indicate the presence of a finite-size plasma jet deflected

tangentially near a parallel magnetic discontinuity.

## 2 Simulation setup

        We consider a three-dimensional simulation geometry that allows the simultaneous investigation of plasma convection along the injection direction, the electrostatic self-polarisation along the normal to the magnetic field direction and the parallel expansion along the magnetic field lines. A schematic overview of the simulation setup is given in Figure 1.

The finite-size plasma jet is illustrated by the red 3-D box. We envisage the case of small Larmor radius plasma jets, i.e., the transversal dimension of the localized plasma structure is much larger than the ion Larmor radius.

        The background magnetic field is parallel to $+Oz$ everywhere inside the simulation domain. Its magnitude increases by 50% between two asymptotic states, $B_1$ – upstream/left of TD and $B_2$ – downstream/right of TD. The thickness of the tangential discontinuity is of the order of $3r_{Le}$ along the $x$-axis (where $r_{Le}$ is the Larmor radius of thermal electrons in the

upstream region). Initially, the electric field is set to zero in the entire simulation domain. We assume that the background steady state magnetic field is established prior to the injection of the plasma jet into the simulation domain and that this





configuration remains unchanged during the entire simulation runtime. The self-consistent magnetic field computed during the simulation runtime adds to the background field. Thus, the simulated plasma jet can be seen as a perturbation to the equilibrium solution describing the TD's magnetic field. Our goal is to investigate the microphysics of the plasma jet – magnetic field interaction process in the vicinity of the tangential discontinuity. The study of TD's formation and stability is

beyond the scope of our paper and shall be considered in our future simulations. Consequently, the background plasma is disregarded in the present work and the finite-size plasma jet is streaming in vacuum towards the magnetic discontinuity, a situation similar to a magnetosheath jet characterized by an excess of dynamic pressure in a rarefied background plasma. The kinetic features discussed in the rest of the paper do not depend on the properties of the background plasma, but on the geometry of the jet and the magnetic and electric field profile.

The electrons and protons forming the 3-D plasma jet are uniformly distributed over a rectangular region localized at the left side of the TD where the magnetic field is uniform. Their initial VDF is a displaced Maxwellian with the average velocity parallel to the positive $x$-axis and perpendicular to the magnetic field direction: $U_0 = 1.2V_{Ti}$ (where $V_{Ti}$ is the thermal velocity of protons). The velocities of both species are initialized according to their corresponding displaced Maxwellian distribution function. Thus, at the beginning of the simulation, the plasma jet is injected into the simulation domain with an

initial bulk velocity pointing towards the discontinuity surface. The jet's plasma has a small beta ($\beta = 0.1$, including both dynamic and thermal plasma pressure) and a large dielectric constant $\varepsilon$. In the simulations discussed in this paper $\varepsilon = 500$, where $\varepsilon = 1 + m_i n/(\varepsilon_0 B^2) = 1 + (\omega_{pi}/\omega_{Li})^2$ is defined as in, e.g., Chen (1974) and Lemaire (1985), with $\omega_{pi}$ the ion plasma frequency and $\omega_{Li}$ the ion Larmor frequency. The ratio between the jet's transversal dimension and the ion Larmor radius is equal to 19. The total simulation time covers $4T_{Li}$ (where $T_{Li}$ is the Larmor period of protons in the upstream region). This simulation

configuration corresponds to the thin, steep and impenetrable TD discussed in detail by Voitcu and Echim (2016, 2017). The PIC code is based on TRISTAN (Buneman, 1993) and has been modified to simulate finite-size plasma structures (jets/plasmoids) interacting with non-uniform magnetic fields. A detailed description of the PIC code is given by Voitcu (2014).

The velocity distribution function, $f$, at different time instances, is computed by using a uniform rectangular grid of

spatial bins that covers the entire plasma jet. For each spatial bin, we calculate histograms by counting the number of particles, $N_t$, in each cell of a uniform rectangular grid defined in the velocity space. Thus, the VDF at time $t$, for a bin of width ($\delta x, \delta y, \delta z$) in configuration space and a cell of width ($\delta v_x, \delta v_y, \delta v_z$) in velocity space, is obtained by dividing $N_t$ to the volume of each 6-D cell in the phase space. The dimensions of the spatial bins and velocity cells are chosen such that the 6-D phase space is well sampled to provide a good representation of the VDF.

The physical quantities are normalized as follows: the number density is normalized to its initial value $n_0$, the particle velocities are normalized to their initial thermal speed $V_{T0} = (2k_B T_0/m)^{1/2}$. The VDF is normalized to the maximum value of the initial displaced Maxwellian distribution, $f_0 = n_0/(\pi^{3/2} V_{T0}^3)$. The time and spatial coordinates are normalized to the initial ion Larmor period $T_{Li}$ and the grid spacing $\Delta x = 0.5 r_{Le}$.

### 3 Numerical results

The dynamics of the jet during its interaction with the TD was discussed by Voitcu and Echim (2016, 2017). When the TD is very steep, the jet is repelled and undergoes two dynamical phases: (A) the jet's frontside interacts directly with the non-uniform magnetic field of the TD leading to vanishing of the self-polarization electric field and (B) the jet is repelled and split into two counterstreaming plasma flows tangential to the TD surface. Here we focus on the kinetic properties of electrons of the deflected plasma jet and consider two instances of its dynamics, typical for phase A (illustrated by Fig. 2)

and phase B (illustrated by Fig. 3), respectively.



We define a uniform grid in configuration space whose cells have the perpendicular size equal to $\delta x=6$ ($\approx 3r_{Le}$) and $\delta y=13$ ($\approx 6r_{Le}$), respectively. This spatial resolution is chosen in order to achieve a sampling of the numerical VDFs comparable to the experimental data obtained with state-of-the-art electron spectrometer in the vicinity of the magnetopause, like, e.g., those embarked on MMS whose time resolution is 30 ms. The distance covered by MMS during 30 ms within a

plasma flow of 100 km/s is of the order of few electron Larmor radii. In order to increase the number of particles collected in each bin of the spatial grid, the grid cells are expanded over the entire column of the simulation domain along the $z$-axis ($\delta z=800$). Nevertheless, we performed tests for smaller $\delta z$ (not shown) and found that the relevant features of the VDF do not depend on the cell size along $Oz$. A uniform 3D grid is also defined in the velocity space whose cells dimensions are $\delta v_z=0.6V_{Te0}$ and $\delta v_x=\delta v_y=0.08V_{Te0}$. In the following we discuss sections of VDFs in the plane perpendicular to the magnetic

field obtained from velocity cells whose centres are located in $v_z=0$.

The numerical simulations results obtained for the interaction of the high-speed plasma jet with the TD are shown in Fig. 2 (for $t=1T_{Li}$, corresponding to phase A) and Figure 3 (for $t=1.9T_{Li}$, corresponding to phase B). In panel (a) we illustrate the number density of electrons for the central cross-section localized in $z=403$; panel (b) shows the 3-D shape of the plasma jet computed as the isosurface corresponding to $n_e=0.04n_0$ in Fig. 2 and $n_e=0.02n_0$ in Fig. 3. The VDF is

calculated for 88 spatial bins that cover the entire jet at $t=1T_{Li}$ and 84 ones at $t=1.9T_{Li}$. We selected the most relevant nine bins labelled A1−A9 in Fig. 2(a) and B1−B9 in Fig. 3(a). The corresponding electron VDFs are shown in panels (c)−(k) of both Fig. 2 and 3. The VDF maps for the entire spatial grids are provided in Appendix A.

The electron VDF in the core of the plasma jet is a displaced Maxwellian similar to the initial one, as can be seen in Fig. 2(c) and 3(c). However, the VDFs at the boundaries of the deflected plasma jet are non-gyrotropic, as shown by bins

A5−A9 and B5−B9 where the phase space density is virtually empty in the central region, for small perpendicular velocities. The VDF takes however larger values in the sector of a ring around the depleted core. These features give the VDF a crescent-like aspect. On the other hand, the VDFs in the frontal bins A2, A3 and B2, B3 are depleted at larger velocities, while the phase space density for small velocities is large.

These non-gyrotropic features of the VDFs are observed in the plane of perpendicular velocities, inside the TD

(bins A5−A7), but also outside it (bins A8, A9 and B5−B9). Their characteristics change with the distance from the jet boundaries. On the one hand, the size of the VDF's central cavity increases towards the lateral edge of the jet (compare, for instance, the VDFs plotted for bins A6 and A7). On the other hand, the VDFs take non-vanishing values only in limited sectors surrounding the central cavity; the peak VDF density in the ring varies from one bin to another (compare, for instance, the VDFs for bins A6 and A8 in Fig. 2).

The physical mechanisms contributing to the formation of crescent-shaped electron VDFs are illustrated by the diagrams in Fig. 4. The electrons in the frontside of the jet interact with the magnetic field gradient inside the discontinuity and drift in the $-Oy$ direction with a velocity proportional to their gyration energy (Northrop, 1963). Thus, the electrons with higher kinetic energies are scattered to larger distances than the less energetic ones. The forward motion of the jet towards the discontinuity continuously replenish the electron population at the front edge. The gradient-B drift acts as a filter of the

perpendicular velocities whose effect is the formation of an energy-dispersed structure in the frontal region of the plasma jet; the electrons with higher kinetic energies accumulate towards the jet's lateral edge, in the direction $-\vec{B}\times\nabla B$ (compare, for instance, Fig. 4(b) and 4(e)). Due to this filtering effect, the regions far from the centre of the jet, along $-Oy$, cannot be reached by the low-energy electrons whose gradient-B drift velocity is small. Thus, far from jet's centre, the phase space of small velocities is empty and a central cavity is formed, as shown in Fig. 4(d) and 4(e). The VDF is therefore ring-shaped.

The corresponding low-energy cut-off of the phase space ring increases with the distance from the jet's centre along $-Oy$, since the spatial regions increasingly farther from the central region of the jet are accessible to electrons with increasingly larger gyration velocities (compare the VDF's central cavities in Fig. 4(d) and 4(e)). Fig. 2 and the diagrams in Fig. 4(a)–4(e) show the formation of ring-shaped VDFs during phase A of the interaction between the jet and the TD.



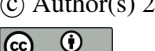

Nevertheless, the kinetic features imprinted by the gradient-B drift during phase A are persistent and the non-Maxwellian electron VDFs are observed later in the simulation when the jet is pushed back, outside the TD. Indeed, during the backward convection of jet's frontal regions, the energy-dispersed structure is displaced in the –*Ox* direction. This explains the shape of VDFs in spatial bins localized in regions of uniform magnetic field, like A8 and A9 in Fig. 2 and

B5−B9 in Fig. 3. Thus, the ring-shaped VDFs persist when the plasma drifts far from the actual interaction with the non-uniform field. The tangential deflection of the jet carries the energy-dispersed structure away from the impact zone and enhances the asymmetry of the lateral edges. The low-energy electrons and the ring-shaped VDFs are displaced in opposite directions, to larger and larger distances from the core. The diagram in Fig. 4(f) illustrates this effect. Note that this mechanism explains partially the simulation results, however it does not account for the non-gyrotropy. To clarify this issue,

we must consider a supplemental effect.

Let us discuss the VDFs obtained in spatial bins A5–A9 and B5–B9 from Fig. 2 and 3, respectively. The diagram in Fig. 4(g) illustrates such a rectangular bin, as well as the guiding centres of four electron orbits localized outside bin's boundaries and whose Larmor radii are comparable to the size of the bin. Each orbit intersects the area of the bin for a limited time interval and is counted when computing the VDF value in the velocity space region determined by the electron

gyration velocity along the segment of the orbit inside the bin. That region in the velocity space changes for each of the four illustrative orbits considered in Fig. 4(g) according to the gyration phase. For instance, within the segment localized in the bin, the gyration velocity of the electron following the red orbit takes values that are restricted to the upper sector of the perpendicular velocity space where $v_y>0$, while the gyration velocity of the electron along the blue orbit is restricted to the $v_x>0$ sector, as shown in Fig. 4(h). The absence of certain guiding centres (and of their intersecting orbits) around the spatial

bin decreases the VDF's phase density for the velocities corresponding to the missing guiding centres. Thus, the spatial inhomogeneity at the borders of the jet manifests as a phase space non-gyrotropy of the distribution function. This effect leads to the formation of non-gyrotropic VDFs in the spatial bin itself and is noticeable in regions of configuration space characterised by sharp variations of the density of particles with large kinetic energy.

The mechanism discussed above can be verified for a configuration similar to our numerical simulations. For this

purpose, a schematic diagram is sketched in Fig. 4(i) where we consider two spatial bins close to jet's boundaries. We discuss the VDFs collected in these two bins. The distribution of the guiding centres around the bins is non-uniform. Indeed, the top bin in Fig. 4(i) is predominantly intersected by large Larmor radius orbits whose guiding centres are found at the left side (like the red orbit in Fig. 4(g)), while the bottom bin is mostly crossed by orbits from above (similarly to the blue orbit in Fig. 4(g)). Therefore, the phase space density of the VDF in the two spatial bins is non-vanishing only in a limited sector

defined by $v_y>0$ (the top bin) and $v_x>0$ (the bottom bin), respectively. Thus, only the upper (right) part of the ring-shaped VDF in Fig. 4(d) (4(e)) is accessible to the top (bottom) bin and the VDFs in these regions are crescent-shaped, as illustrated in Fig. 4(j) and 4(k). This explains the results of our simulations. Indeed, if one checks the electron VDFs from the numerical simulations in Fig. 2, one notes the crescent-like features in bins A6 and A8 in Fig. 2(a) similar to the top and bottom bins in Fig. 4(i) (compare the VDFs in Fig. 2(h) and 2(j) to the ones in Fig. 4(j) and 4(k)). On the other hand, bin B5 in Fig. 3(a) is

crossed by orbits having the guiding centres localized at its left side and also above it. The crescent-shaped VDF in this case is rotated towards the ($v_x>0$, $v_y>0$) sector of the perpendicular velocity space, as can be seen in Fig. 3(g). Similar arguments explain the orientation of all crescent-shaped VDFs in Fig. 2 and 3.

The discussion above shows that the formation of crescent-like VDFs at the edges of a plasma jet interacting with a magnetic discontinuity is the result of a combined effect due to: (i) the formation of an energy dispersion region at the front

edge of the plasma jet sustained by the energy-dependent gradient-B drift – this effect leads to the formation of a ring shaped VDF and (ii) the remote sensing of large Larmor radius electrons at the lateral edge of the jet – this effect explains why certain regions of the ring-shaped VDFs have very small phase density leading to the crescent-like shape. The simulations also show that the kinetic structure resulting from the interaction with the discontinuity is maintained at later stages, when





the jet is deflected and pushed outside the TD. Indeed, the non-gyrotropic features imprinted by the interaction of the plasma jet with the steep tangential discontinuity are observed during its subsequent motion.

Numerous recent spacecraft observations show the omnipresence of magnetosheath jets and therefore there is an increased interest on their kinetic structure (see, e.g., Plaschke et al., 2017; Karlsson et al., 2018). Below we discuss a
possible method to check the results of our simulations against spacecraft observations. For this purpose, we "launched" a virtual satellite into the simulation domain. The VDF is "measured" by counting the particles localized within a fixed spatial bin centred on the virtual satellite. We use the same spatial resolution as in Fig. 2 and 3, thus the resolution of the virtual satellite measurements is comparable to the very high-resolution electron measurements provided by MMS. In Fig. 5 we show five examples of electron VDFs measured by the virtual satellite. For consistency, we consider the same satellite S1 as
in Voitcu and Echim (2017). A virtual time-series of VDFs is collected in $x=75$, $y=153$ and $z=403$, at $t_1=1.03T_{Li}$, $t_2=1.54T_{Li}$, $t_3=2.05T_{Li}$, $t_4=2.57T_{Li}$ and $t_5=3.08T_{Li}$. We illustrate the perpendicular cross-sections corresponding to $v_z=0$ and use the same resolution in velocity space as in Fig. 2 and 3. The temporal separation of the five VDFs is equal to $19T_{Le}$ (where $T_{Le}$ is the Larmor period of electrons in the upstream region); it corresponds to a 30 ms time resolution which is compatible with the MMS measurements of electron VDFs in a magnetic field of 23 nT.

The virtual satellite probes the jet's lateral edge deflected towards the $-\vec{B} \times \nabla B$ direction in the vicinity of the TD (which could correspond to an idealized magnetopause). It moves within the energy-dispersed structure and detects the crescent-shaped electron VDFs illustrated in Fig. 5(a)−5(e). The morphology and orientation of the electron VDFs change in time. At $t_1=1.03T_{Li}$ (see Fig. 5(a)), the crescent is thinner and the phase density is enhanced in the $v_x>0$ sector of the perpendicular velocity space (as in Fig. 2(k)). Up to $t_5=3.08T_{Li}$ (see Fig. 5(e)), the crescent becomes more and more thicker
and it rotates counter clockwise towards the $v_y>0$ sector, as the virtual satellite samples different areas close to the jet's boundaries. In a frame of reference attached to the deflected plasma jet, the satellite is moving away from the jet's edge at the lowest $y$-value, drifting along the positive direction of the $y$-axis and in the same time closer to the TD's surface. Indeed, at time $t_1$, the area sampled by the virtual satellite is similar to that of bin A9 in Fig. 2(a), while at time $t_5$ the satellite is in a similar position to the centre of the bin A5 in Fig. 2(a). Thus, the position of the satellite changes with respect to the plasma
jet's core. At time $t_1$, the satellite is at its farthest distance from the jet's core along the $-\vec{B} \times \nabla B$ direction and the corresponding VDF has the largest central cavity in the space of perpendicular velocities. Later on, the satellite moves closer to the jet's core and the size of the VDF's central cavity decreases. The position of the satellite changes also with respect to the local orientation of the jet's boundaries. While at time $t_1$ most of the electrons are coming from above the satellite's sampling area (like the blue orbit in Fig. 4(g)), at $t_5$ most of them are coming from the left side (like the red orbit in Fig.
4(g)). Consequently, the peak of the gyrophase density in the space of perpendicular velocities rotates from the $v_x>0$ sector in Fig. 5(a) towards the $v_y>0$ sector in Fig. 5(e). Since the properties of the measured electron VDFs depend on the relative location of the satellite with respect to the jet's core and boundaries, such kind of velocity distribution functions could be used to estimate some geometrical and morphological features of a deflected plasma jet.

## 4 Summary and conclusions

3-D PIC simulations of the interaction between a low-beta plasma jet and a thin, steep and impenetrable tangential discontinuity with no magnetic shear emphasize the formation of crescent-shaped electron velocity distribution functions at the jet's boundaries. The simulation setup corresponds to an idealised, yet relevant, magnetic configuration likely to be observed at the frontside magnetopause during northward IMF. The results of our simulations show:

1. The crescent-shaped electron VDFs are observed in a magnetic field configuration that has no X-line. Their occurrence
40       and characteristics depend on the dynamic and geometric properties of the jet, as well as on the profile of the non-uniform magnetic and electric fields at the discontinuity.



2. The energy-dependent gradient-B drift of the electrons in the frontside of the jet leads to the formation of an energy-dispersed structure with ring-shaped velocity distribution functions: the electrons with higher gyration energies accumulate towards the jet's lateral edge along the $-\vec{B} \times \nabla B$ direction where the VDF shows a deficit of low energy electrons in the perpendicular velocity space.

3. Within the most lateral edges of the energy-dispersed structure, the ring of the VDF is incomplete and the VDF's shape changes from ring-shaped to crescent-shaped. The crescent is due to the spatial anisotropy at the sharp edges of the plasma jet. It is produced by the remote sensing of high-energy particles whose guiding centres are localized inside the jet, as illustrated schematically in Fig. 4.

4. The crescent-shaped VDFs are produced simultaneously by (i) the velocity filtering effect of the gradient-B drift and (ii) the remote sensing of large Larmor radius electrons at the boundaries of the plasma jet.

5. Virtual time-series of electron VDFs recorded by a virtual satellite launched into the simulation domain show that the ring-shaped and crescent-shaped electron VDFs are persistent and carried by the jet during its motion after interacting with the magnetic discontinuity. Their properties (like the size and position of the crescent) change with the satellite's location within the jet, thus such VDFs can help estimating the geometrical and morphological features of a deflected jet during or after its interaction with a tangential discontinuity or regions of sharp magnetic field variation.

The new and original result of our study is the identification of crescent-shaped electron VDFs as a boundary effect at the edges of a finite-size three-dimensional plasma jet impacting on a tangential discontinuity with no field reversal. The properties of the crescents vary with the distance from the jet's centre in the $-\vec{B} \times \nabla B$ direction; they also depend on the local orientation of the jet's boundaries. The physical mechanism we propose to explain the crescent-shaped VDFs is based on finite Larmor radius effects. To our knowledge, this is for the first time when such kind of electron velocity distributions are emphasized in the absence of X-lines and magnetic field reversal and the physical process proposed adds to the ones discussed in the past like, for instance, the remote sensing of a thin Harris sheet (Lee et al., 2004; Voitcu and Echim, 2012), the dynamics of non-magnetized Speiser-like electrons (Bessho et al., 2016; Shay et al., 2016; Lapenta et al., 2017) or the dynamics of magnetized electrons (Egedal et al., 2016) in the diffusion region of magnetic reconnection sites.

Our simulations provide new insight on the microphysics of plasma-field interaction near magnetic boundaries and contribute to advance the understanding of the dynamics and kinetics of magnetosheath high-speed jets streaming towards the magnetopause during periods of northward IMF. We also demonstrate that the crescent is a persistent feature of the VDFs that can be detected at relatively large distances from the interaction with the discontinuity and therefore might be used as a kinetic signature of such interaction.

**Data availability**

The simulation data used to produce all the plots and analyses included in this paper can be requested by sending an e-mail to Gabriel Voitcu at one of the following addresses: gabi@spacescience.ro or gabriel.voitcu@gmail.com.

**Appendix A**

We present here the full map of the electron velocity distribution functions for all the spatial bins defined in the $xOy$ plane, perpendicular to the magnetic field direction. In Fig. A1 we show the electron number density in the plane $xOy$ for $z=403$, at two instances: (a) $t=1T_{Li}$ and (b) $t=1.9T_{Li}$, together with all the bins used to calculate the electron VDFs. There are 88 bins in Fig. A1(a) and 84 bins in Fig. A1(b) that cover the entire plasma jet at the two different time instances. The VDFs for each spatial bin are illustrated in Fig. A2 (at $t=1T_{Li}$) and A3 (at $t=1.9T_{Li}$) where we show cross-sections corresponding to $v_z=0$. The nine selected bins A1–A9 in Fig. 2 of the article correspond to bins 39, 10, 21, 43, 65, 76, 87, 75, 86 in Fig. A1(a).

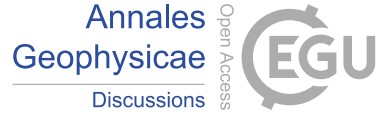



The nine selected bins B1–B9 in Fig. 3 of the article correspond to bins 39, 6, 13, 41, 69, 76, 83, 75, 82 in Fig. A1(b). One note that in the bulk of the jet the VDF is a displaced Maxwellian, consistent with the forward motion of the plasma. The non-gyrotropic effects are localized at the edges of the plasma jet.

**Competing interests**

The authors declare that they have no conflict of interest.

**Acknowledgments**

The authors acknowledge support from the Romanian Space Agency (ROSA) through the Space Technology and Advanced Research (STAR) Programme under contracts 182/2017 (OANA) and 122/2017 (ODYN) and also from the Romanian Executive Agency for Higher Education, Research, Development and Innovation Funding (UEFISCDI) through project

VESS/2018. Marius Echim acknowledges support from the Belgian Solar-Terrestrial Centre of Excellence (STCE) in Brussels, Belgium.

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





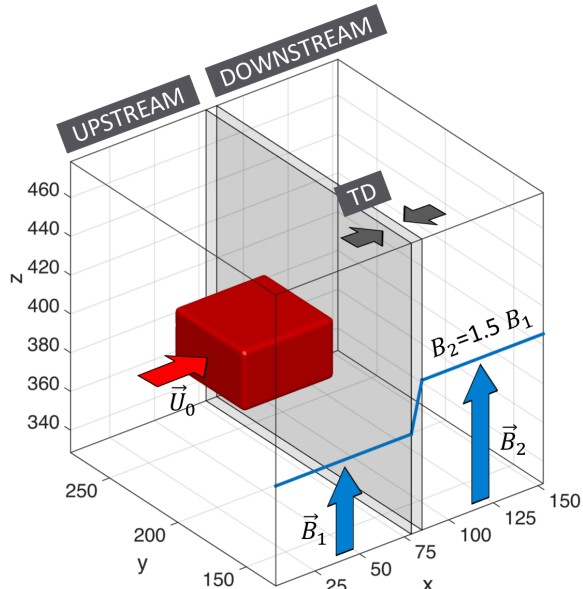

**Figure 1: Schematic diagram of the simulation setup. The high-speed plasma jet (shown by the red 3-D box) moves from the upstream region towards a parallel tangential discontinuity. The spatial profile of the background magnetic field is shown with blue. The tangential discontinuity is located between $x_1=79$ and $x_2=85$.**





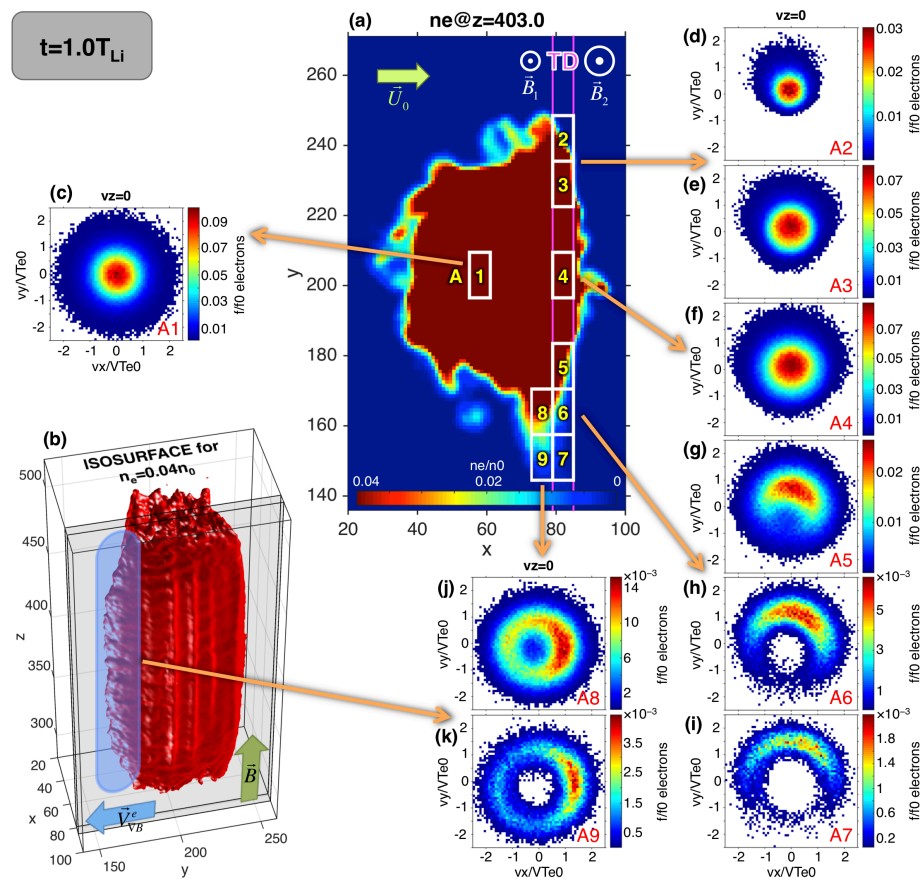

**Figure 2:** Snapshot of 3-D PIC simulation at $t=1T_{Li}$ (phase A). (a) Electron number density in the *xOy* plane for *z*=403. (b) 3D density isosurface for $n_e=0.04n_0$. (c)−(k) Electron VDFs in the space of perpendicular velocities for the spatial bins shown with white rectangles in panel (a). The two magenta lines mark the TD's boundaries.





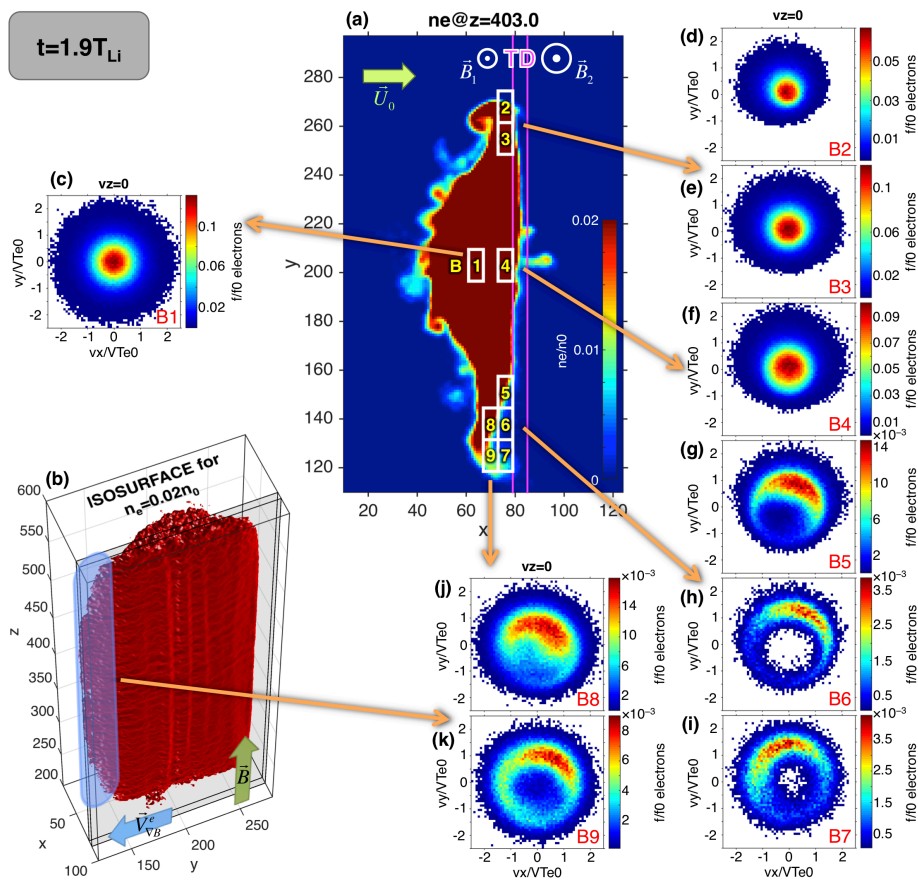

**Figure 3: Snapshot of 3-D PIC simulation at $t$=1.9$T_{Li}$ (phase B). (a) Electron number density in the $xOy$ plane for $z$=403. (b) 3D density isosurface for $n_e$=0.02$n_0$. (c)–(k) Electron VDFs in the space of perpendicular velocities for the spatial bins shown with white rectangles in panel (a). The two magenta lines mark the TD's boundaries.**



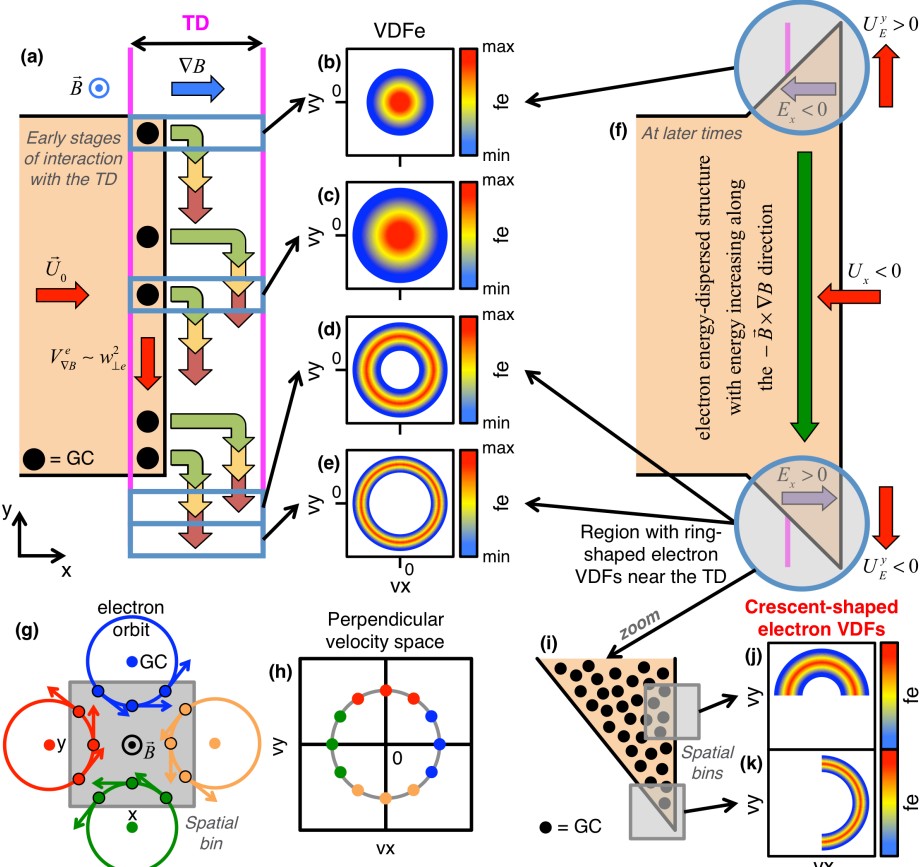

**Figure 4: Panel (a) illustrates the phase A of plasma jet's interaction with the magnetic discontinuity. The black dots indicate the guiding centres (GCs) of electrons localized in the front edge. The multiple groups of green-yellow-red arrows show the gradient-B drift displacement of the GCs proportional to the gyration energy (green corresponds to the lowest energy). The four blue boxes indicate the bins where the VDF is computed. The electron VDFs obtained for the upper edge bins (panels (b) and (c)) are gyrotropic and their phase density vanishes for large velocities; the VDFs obtained for the bottom edge bins (panels (d) and (e)) are ring-shaped. Panels (g) and (h) illustrate how the electrons with GCs outside a spatial bin contribute to different sectors of the VDF collected in the respective bin. The colours of the dots in panel (h) identify which particle shown in panel (g) contributes to that sector of the VDF. This effect corresponds to a remote sensing of the large Larmor radius electrons outside the bin. Panels (f), (i)−(k) show how the remote sensing effect contributes to the formation of crescent-shaped VDFs in a configuration similar to the PIC simulations.**





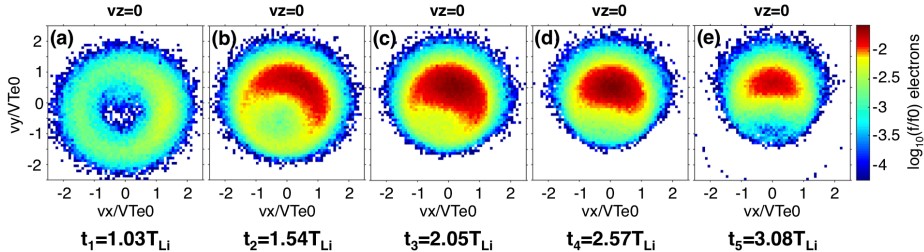

**Figure 5:** Virtual time-series of five electron VDFs "measured" by a virtual satellite launched into the simulation domain. We illustrate the space of perpendicular velocities for $v_z=0$. The colour scale used is logarithmic.





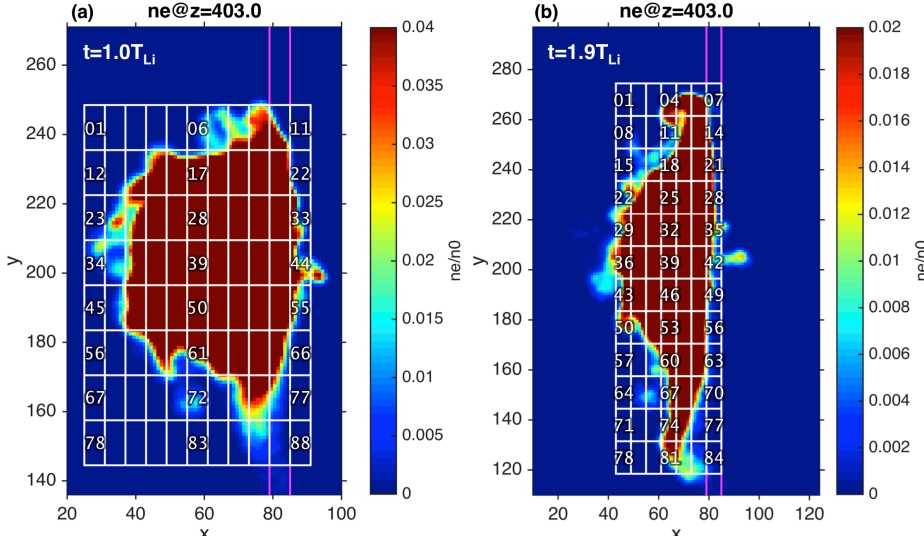

**Figure A1: Electron number density for *z*=403 cross-section together with the entire spatial grid used to calculate the electron velocity distribution functions, at (a) *t*=1*T*$_{Li}$ and (b) *t*=1.9*T*$_{Li}$. There are 88 spatial bins in panel (a) and 84 ones in panel (b).**

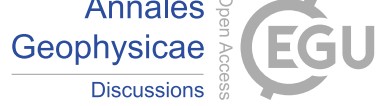



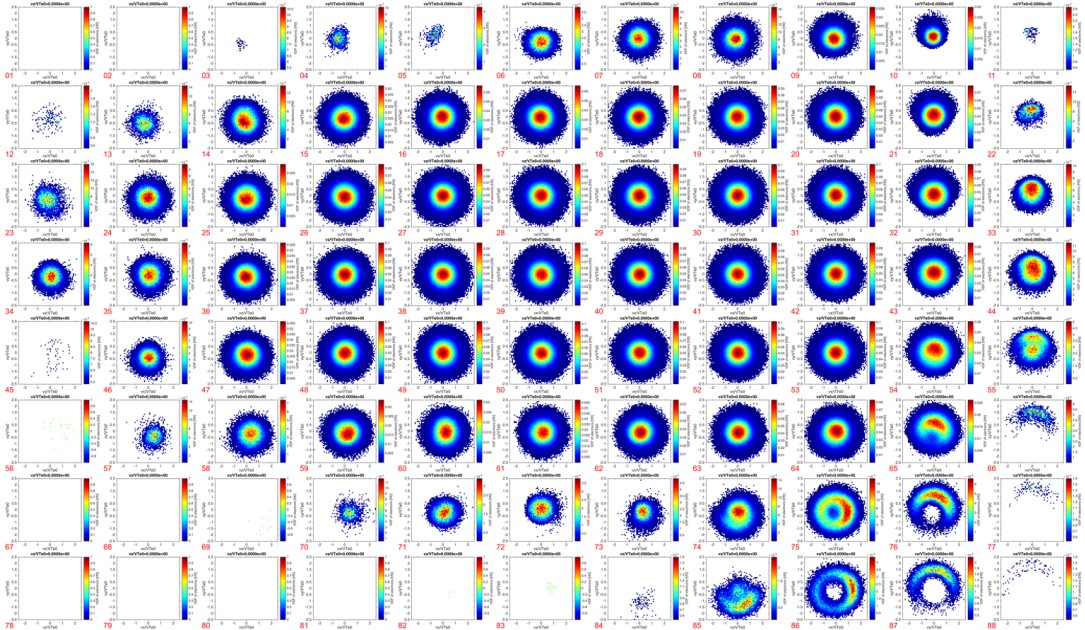

**Figure A2: Electron VDFs, at $t=1T_{Li}$, for each of the 88 spatial bins defined in Figure A1(a). We show $v_z=0$ cross-sections. The nine VDFs presented in Figure 2 of the article correspond to bins 39, 10, 21, 43, 65, 76, 87, 75 and 86.**





**Figure A3: Electron VDFs, at $t$=1.9$T_{Li}$, for each of the 84 spatial bins defined in Figure A1(b). We show $v_z$=0 cross-sections. The nine VDFs presented in Figure 3 of the article correspond to bins 39, 6, 13, 41, 69, 76, 83, 75 and 82.**