# Peer review of "Crescent-shaped electron velocity distribution functions formed at the edges of plasma jets interacting with a tangential discontinuity"

_Annales Geophysicae, 2018_

## Referee Comment (RC1) · Anonymous Referee #1 · 7 Oct 2018

General comments:

The manuscript "Crescent-shaped electron velocity distribution functions formed at the edges of plasma jets interacting with a tangential discontinuity" by G. Voitcu and M. Echim discusses a mechanism explaining the formation of crescent-shaped electron distribution functions by means of fully-kinetic 3D PIC simulations of plasma clouds deflected by tangential discontinuities.

The manuscript is well written and clear. It shows an interesting and physically clear

explanation for the now commonly observed crescent shaped electron distribution functions in the Earth's magnetosphere (in particular near magnetic reconnection regions). The Figure 4 with the schematics of the mechanism is particularly helpful, emphasizing the role of the filtering of high vs low energy electrons by the gradient-B drift plus a remote sensing effect in magnetic gradients (without field reversal as in typical magnetic reconnection geometries). I also appreciate the details given for the calculation of the distribution functions (precise location) and virtual satellites. As I far as I know, I was not aware of other works pointing out to this mechanism for crescent-shaped (or non-gyrotropic in general) electrons distribution functions in this scenario. I do not see any major problem with the interpretation of the results of the PIC simulations. I would only ask the authors for some clarifications and additional details about the initial setup of those simulations, in order to guarantee reproducibility of the results.

Specific comments:

1) This manuscript points out to two earlier papers [Voitcu2016], [Voitcu2017] for the description of the simulation setup. But in order to be more self-contained, I would ask the authors to add some important parameters to the text. In addition, I could not find anywhere the values of other critical quantities (or at least it was not clear to me where to find them or calculate them). I think the manuscript and its readers will benefit of those details. i) ion or electron plasma beta of the plasma jet including only thermal pressure (not the bulk velocity). Or equivalently, the value of the electron thermal speed. The value of eta=500 allows to obtain omega_pi/omega_ci, but the ion (or electron) thermal speed are still needed to determine other quantities. ii) What are the boundary conditions? (I know that this is mentioned in previous papers, but nothing here) iii) How far away are the boundaries with respect to the plasma cloud and/or tangential discontinuities? Table 1 of [Voitcu2016] is not clear regarding this: the simulation box is expressed in number of grid points, but it is more helpful to convert it to electron gyroradius or other physical units (the lengths in all the figures should also be normalized to some meaningful physical units, if possible). iv) What is the number

of particles per cell and/or the total number of particles?

2) One parameter that I am puzzled about is the grid resolution. Table 1 of [Voitcu2016] says that Delta x/lambda_De = 2.5. Is it exactly the same for this paper? If so, that would indicate an under-resolved grid cell size, possibly leading to numerical heating. Is there any indication of that effect? How well conserved is the total energy in those simulations? Have the authors used some high order shape function or some kind of current smoothing to prevent numerical heating? And how well resolved is the electron gyroradius compared to the grid cell size? (this quantity should also be well resolved).

3) I understand that the proposed mechanism for the formation of crescent shaped electrons is based on the remote sensing of electrons with an electron gyroradius on the order of the magnetic field gradient in a tangential discontinuity. As mentioned in the references of this manuscript, similar mechanisms were proposed for ions, which have a much larger gyroradii and therefore they are much easier to detect. Is there any chance to actually measure in-situ such a steep magnetic field gradients? (in the Earth magnetosphere, for example by MMS). I am not sure about that, in particular considering the low-beta plasma used in this simulation study, which implies a very small electron gyroradii. I would appreciate if the authors could add some studies with explicit numbers for measured tangential discontinuities (and corresponding electron gyroradii), please.

4) Related to the previous question: How about high-beta plasma effects? In this parameter regime, the electron gyroradii would be larger and easier to measure. Would there be any significant different for the proposed mechanism in that parameter regime? Have the authors tried such simulations? If not, why not? In such cases electrostatic effects such as polarization electric fields should be diminished. I think a brief discussion about this issue will benefit the readers by making the results of this manuscript more general and with a wider applicability.

---

## Referee Comment (RC2) · Anonymous Referee #2 · 9 Oct 2018

This paper desctibes PIC-simulations of a magnetosheath jet (i.e. a localized increase in dynamic pressure caused by increases of density, velocity or a combination of both), crossing a northward IMF, dayside magnetopause (modelled as a tangential discontinuity without shear). The authors present a case where the jet is not penetrating the magnetopause, and in particular present detailed electron distribution functions from different parts of the jet at different stages of its interaction with the magnetopause.

The paper is well written with clear figures and results. The code and methodology builds on earlier studies and appears to be well tested. The results are clear and

interesting. In my opinion, the paper is almost ready for publications. I only have a few minor issues, listed below:

Minor comments:

1) What is the relevance of the number of the dielectric constant given on page 5, line 13? It is not commented further in the paper.

2) The authors note themselves that crescent-shaped electron distributions are also found in connection with X-lines and reconnection regions at the dayside magnetopause. It would be helpful if the authors discussed in more detail similarities and differences between those distributions and the ones found in this paper. Is it possible to differentiate by the two cases by just inspecting the distribution functions?

3) I think that the authors should discuss the relevance of their results a little bit more. They mention that the distribution functions can be used to estimate the geometry of jets, but there should be further points to discuss. For example, I assume that the resulting distribution functions are unstable to wave generation. What was could be expected to be excited? And how would moment calculations, which assume Maxwellian distribution, be affected?

4) No ion distributions are shown in this paper. The authors could at least comment on if any effects of the interaction are shown in those.
* * *

---

## Author Comment (AC1) · 23 Oct 2018

The comment was uploaded in the form of a supplement:
https://www.ann-geophys-discuss.net/angeo-2018-102/angeo-2018-102-AC1-supplement.pdf

---

## Author Comment (AC2) · 23 Oct 2018

**REPLY TO REVIEWER #2 COMMENTS**

We thank the referee for the recommendations to improve the manuscript. In the following we provide a response to all items included in the referee report. We hope that the changes made in the text will render it more comprehensive. The new added text is shown in red in the marked-up version of the manuscript.

*Reviewer comments*

**1.** *What is the relevance of the number of the dielectric constant given on page 5, line 13? It is not commented further in the paper.*

**Answer:**

The dielectric constant is relevant in the context of plasma self-polarization. This is the actual physical mechanism which sustains the propagation of the plasma jet across the transverse magnetic field in our simulations. For plasmas with $\varepsilon \gg 1$, as space plasmas for instance, collective dielectric effects are important, as discussed by *Schmidt* (1960) and *Lemaire* (1985). In this case, a polarization electric field is formed inside the localized plasma structure and enables the further convection of the jet across the perpendicular magnetic field (see *Voitcu and Echim*, 2016 for a detailed discussion on this topic in the context of our simulations). To render more clearly this aspect into our manuscript, an explanatory text has been added in Section 2 (pg. 3, lines 21-23).

**2.** *The authors note themselves that crescent-shaped electron distributions are also found in connection with X-lines and reconnection regions at the dayside magnetopause. It would be helpful if the authors discussed in more detail similarities and differences between those distributions and the ones found in this paper. Is it possible to differentiate by the two cases by just inspecting the distribution functions?*

**Answer:**

The main differences between the crescent-shaped electron VDFs obtained in our simulations and those reported in connection with magnetic reconnection sites at the dayside magnetopause are related to (i) the different physical mechanisms leading to their formation and also with (ii) the different spatial geometry involved in those processes.

The physical mechanisms proposed by, e.g., *Bessho et al.* (2016), *Shay et al.* (2016) or *Lapenta et al.* (2017), for formation of crescent-shaped electron VDFs involve meandering electron orbits in the vicinity of X-lines, thus in spatial regions characterized by a magnetic field reversal. In a similar geometry, *Egedal et al.* (2016) proposed that the perpendicular crescent-shaped distributions are produced by the diamagnetic drift of well-magnetized electrons in a thin boundary layer localized along the magnetopause separatrix. In our case, the physical mechanism we propose does not involve the spatial reversal of the magnetic field, since we consider a parallel tangential discontinuity. The crescent-shaped electron VDFs obtained in our simulations are formed by two combined effects: (i) the velocity filtering effect produced by the grad-B drift and (ii) the remote sensing effect of large Larmor radius electrons.

The electron crescents reported here are obtained in connection to a jetting plasma structure that moves in opposite directions tangentially to the parallel discontinuity surface

(our simulated magnetopause) and perpendicular to the magnetic field; they are found inside and also in the vicinity of the discontinuity towards the $-\vec{B} \times \nabla B$ direction (which is the $-\vec{M}$ direction in the (L,M,N) boundary-normal coordinates that are used, for instance, in *Burch et al.* (2016)). On the other hand, the electron crescents reported by *Burch et al.* (2016) have been observed in connection to the anti-parallel jetting along the $\vec{L}$ direction produced by magnetic reconnection at the dayside magnetopause; they are found close to the magnetic null region and along the $\vec{N}$ direction, as obtained also by, e.g., *Bessho et al.* (2016) and *Lapenta et al.* (2017). In the study of *Egedal et al.* (2016), the electron crescents are obtained in a boundary layer having the width of ~$1 r_{Le}$ all along the low-density separatrix. On the contrary, in our simulations, the electron crescents are obtained over a region in the perpendicular plane to the magnetic field that covers ~$12 r_{Le}$ along *Ox* (the $\vec{N}$ direction) and ~$20\text{-}30 r_{Le}$ along *Oy* (the $\vec{M}$ direction) (see Figures A2 and A3 in Appendix A of the manuscript). Also, the low-energy cut-off in the perpendicular velocity space reported by *Egedal et al.* (2016) is related to an exclusion perpendicular energy that varies with distance from the separatrix along the $\vec{N}$ direction, while in our simulations the low-energy cut-off varies with distance from the jet's centre along the $\vec{M}$ direction.

As emphasized in the concluding remarks of our manuscript: *"the crescent is a persistent feature of the VDFs that can be detected at relatively large distances from the interaction with the discontinuity and therefore might be used as a kinetic signature of such interaction"*. Thus, the crescent-shaped electron VDFs can be used to identify plasma jets that have interacted directly with a magnetic discontinuity, but additional information is needed as, for instance, data about the magnetic field, particle density or bulk velocity. A simple inspection of only crescents cannot discriminate between the different physical mechanisms leading to their formation.

To render more clearly this aspect into our manuscript, an explanatory text has been added in Section 4 (pg. 7, lines 36-40 & pg. 8, lines 1-8).

**3.** *I think that the authors should discuss the relevance of their results a little bit more. They mention that the distribution functions can be used to estimate the geometry of jets, but there should be further points to discuss. For example, I assume that the resulting distribution functions are unstable to wave generation. What was could be expected to be excited? And how would moment calculations, which assume Maxwellian distribution, be affected?*

**Answer:**

The velocity distribution functions obtained in our simulations are, indeed, unstable to wave generation. The ring-shaped electron VDFs produced by the velocity-filtering effect can generate, for instance, cyclotron harmonic waves (e.g. *Tataronis and Crawford*, 1970; *Lee and Birdsall*, 1978). It should be mentioned that the study of waves generation and instabilities in this simulation is beyond the scope of the present paper and shall be addressed in our future studies. Thus, it might be premature at the present time to approach this aspect into the manuscript.

Under certain circumstances, the moment calculation can be affected by the shape of the velocity distribution function. It has been shown by *Lee et al.* (2004) that the moment computation for non-gyrotropic (or crescent) VDFs may provide unrealistically large bulk velocities. Indeed, in such cases of non-Maxwellian distributions, large velocity moments can

be obtained even for a stationary plasma, if those distributions are produced by a remote sensing effect, as the one discussed also in this paper. Therefore, when calculating the moments of this kind of velocity distribution functions, both the distributions and moments should be examined carefully to establish if the large velocity moments really indicate an actual plasma flow or not.

**4.** *No ion distributions are shown in this paper. The authors could at least comment on if any effects of the interaction are shown in those.*

**Answer:**

The crescent-shaped VDFs are observed only for electrons in the current simulations, while the ion VDFs show no crescent features. Indeed, for the particular set of parameters used here, the ion grad-B drift velocity is smaller than for electrons and the corresponding velocity filtering effect is not large enough to produce an efficient scattering of ions along the $+\vec{B} \times \nabla B$ direction leading to an ion energy-dispersed structure with crescent-shaped ion distributions. Additional simulations are envisaged in the future to study the formation of crescent-shaped ion VDFs simultaneously with the electron crescents. To render more clearly this aspect into our manuscript, an explanatory text has been added in Section 3 (pg. 6, lines 14-15).

**Crescent-shaped electron velocity distribution functions formed at the edges of plasma jets interacting with a tangential discontinuity**

Gabriel Voitcu[1], Marius Echim[1,2]

[1]Institute of Space Science, Magurele, 077125, Romania
5  [2]Belgian Institute of Space Aeronomy, Brussels, B-1180, Belgium
*Correspondence to*: Gabriel Voitcu (gabi@spacescience.ro)

**BLUE** text = new text in response to **Referee #1** comments
**RED** text = new text in response to **Referee #2** comments

[revised manuscript text omitted]

---

## Referee Comment (RC3) · Anonymous Referee #2 · 24 Oct 2018

The authors have addressed all my issues, and I can now recommend the paper for publication.

---

## Author Comment (AC3) · 26 Oct 2018

We thank the referee for reviewing our manuscript.

---

## Author Response (AR1)

We thank the referee for the recommendations to improve the manuscript. In the following we provide a response to all items included in the referee report. We hope that the changes made in the text will render it more comprehensive. The new added text is shown in blue in the marked-up version of the manuscript.

**Reviewer comments**

**1.** *This manuscript points out to two earlier papers [Voitcu2016], [Voitcu2017] for the description of the simulation setup. But in order to be more self-contained, I would ask the authors to add some important parameters to the text. In addition, I could not find anywhere the values of other critical quantities (or at least it was not clear to me where to find them or calculate them). I think the manuscript and its readers will benefit of those details. i) ion or electron plasma beta of the plasma jet including only thermal pressure (not the bulk velocity). Or equivalently, the value of the electron thermal speed. The value of eta=500 allows to obtain omega_pi/omega_ci, but the ion (or electron) thermal speed are still needed to determine other quantities. ii) What are the boundary conditions? (I know that this is mentioned in previous papers, but nothing here) iii) How far away are the boundaries with respect to the plasma cloud and/or tangential discontinuities? Table 1 of [Voitcu2016] is not clear regarding this: the simulation box is expressed in number of grid points, but it is more helpful to convert it to electron gyroradius or other physical units (the lengths in all the figures should also be normalized to some meaningful physical units, if possible). iv) What is the number of particles per cell and/or the total number of particles?*

**Answer:**
The second section of the article ("*Simulation setup*") has been modified to include the referee's requests regarding: **(i)** the electron beta-parameter computed without the dynamic pressure [$\beta_e$=0.09, pg. 3, lines 18-19] and the electron-to-ion thermal speed ratio [$V_{Te}/V_{Ti}$=30, pg. 3, line 14], **(ii)** the boundary conditions [periodic for both particles and fields, pg. 3, lines 28-32], **(iii)** the size of the simulation box [155×405×805 grid cells, pg. 3, line 17] and plasma cloud [50×50×30 grid cells, pg. 3, line 17] and also the position of the tangential discontinuity [from $x_1$=79 to $x_2$=85, pg. 2, line 39], **(iv)** the number of particles used in the simulation (total [30 millions for each species] and per cell [400 for each species], pg. 3, lines 30-32). For consistency with our previous two articles on this topic (*Voitcu and Echim*, 2016, 2017), we decided to keep the same units for the spatial coordinates which are normalized to the grid spacing $\Delta x$. To convert the units to electron gyroradii, we provide the ratio between $\Delta x$ and the gyroradius of thermal electrons in the injection region: $\Delta x/r_{Le}$=0.5.

**2.** *One parameter that I am puzzled about is the grid resolution. Table 1 of [Voitcu2016] says that Delta x/lambda_De = 2.5. Is it exactly the same for this paper? If so, that would indicate an under-resolved grid cell size, possibly leading to numerical heating. Is there any indication of that effect? How well conserved is the total energy in those simulations? Have the authors used some high order shape function or some kind of current smoothing to prevent numerical heating? And how well resolved is the electron gyroradius compared to the grid cell size? (this quantity should also be well resolved).*

**Answer:**

The ratio between the grid spacing and the electron Debye length is indeed equal to $\Delta x/\lambda_{De}$=2.5, as in our previous simulations (*Voitcu and Echim*, 2016, 2017). The time step and grid spacing used to run the simulations have been carefully selected to avoid the numerical instabilities related to the discretization of space and time that could lead to unphysical effects, as the numerical heating mentioned by the referee. According to *Cai et al.* (2010; section 2.11, page 39) and *Birdsall and Langdon* (1991; section 8-13, page 179), for a linear weighting scheme (as the one used in our simulations), the following interval:

$$0.1 \leq \frac{\lambda_{De}}{\Delta x} \leq 0.3$$

must be avoided when selecting the spatial resolution of the simulation. In our case, the ratio between the electron Debye length and the grid spacing is $\lambda_{De}/\Delta x$=0.4, thus outside the aforementioned forbidden interval. Moreover, we apply a smoothing procedure to the current density in order to improve the overall accuracy of the simulation and reduce the numerical noise that could cause unwanted heating. For this purpose, the 3-D version of the three-point binomial filter along each direction in configuration space is used, leading to 27 weights that are applied to smooth the current density (see *Birdsall and Langdon*, 1991; appendix C, page 437, for the 1-D and 2-D cases):

$$\vec{J}_s(i,j,k) = \frac{M \cdot \vec{J}(i,j,k) + S \cdot S_{terms} + K \cdot K_{terms} + C \cdot C_{terms}}{M + 6S + 12K + 8C}$$

where $S_{terms}$, $K_{terms}$ and $C_{terms}$ correspond to the values of $\vec{J}$ in 26 neighboring nodes around ($i$, $j$, $k$), while $M$=8, $S$=4, $K$=2 and $C$=1. This smoothed current density is further used in Ampère's law to obtain the self-consistent electric field.

In Figure R1 we illustrate the time history of the total energy corresponding to our simulation (kinetic + electric + magnetic). We show the deviation of the total energy from its initial value. As can be noticed, there is a very small increase of the total energy during $4T_{Li}$ of runtime. Indeed, the asymptotic value of $\Delta W/W_0$=0.0016% (the red line in Figure R1), while the maximum deviation during the initial stages of the simulation is smaller than 0.004%. This small increase of the total energy is not forbidden for momentum conserving codes (as ours) which are not strictly energy conserving. Thus, we consider that the total energy in sufficiently well conserved in our simulations.

[Figure]

**Figure R1.** Time history of the total energy $W$ of the simulated system (kinetic + electric + magnetic) during four ion Larmor periods of runtime. $W_0$ is the initial energy.

We consider that the electron gyroradius is sufficiently well resolved by the spatial grid used in our simulations. The ratio between the gyroradius of the thermal electrons in the injection region and the grid spacing is equal to $r_{Le}/\Delta x = 2.1$. Thus, the Larmor orbit of such an electron is covered by 25 grid nodes in the perpendicular plane to the magnetic field. Since we are using the linear weighting for the shape function, a linear grid-to-particle interpolation scheme is also used to compute the fields in the position of each particle. In Figure R2 we illustrate the trajectory of an electron extracted from the actual simulation used in this manuscript. We show a single electron Larmor period. The gyration velocity is equal to $0.97V_{Te}$. As can be noticed, the orbit is well resolved and the Larmor radius has the correct value. Note the zero-order electric drift along $+Ox$. Also, there is a small grad-B drift along $-Oy$ due to some small variations of the magnetic field around the background value at the left side of the tangential discontinuity.

[Figure]

**Figure R2.** The trajectory of a thermal electron at the left side of the tangential discontinuity during a single gyration period. The red point marks the initial position of the electron, while the black dots show the grid points where the fields are computed.

**3.** *I understand that the proposed mechanism for the formation of crescent shaped electrons is based on the remote sensing of electrons with an electron gyroradius on the order of the magnetic field gradient in a tangential discontinuity. As mentioned in the references of this manuscript, similar mechanisms were proposed for ions, which have a much larger gyroradii and therefore they are much easier to detect. Is there any chance to actually measure in-situ such a steep magnetic field gradients? (in the Earth magnetosphere, for example by MMS). I am not sure about that, in particular considering the low-beta plasma used in this simulation study, which implies a very small electron gyroradii. I would appreciate if the authors could add some studies with explicit numbers for measured tangential discontinuities (and corresponding electron gyroradii), please.*

**Answer:**
The mechanism we propose for the formation of crescent-shaped electron velocity distribution functions does not require tangential discontinuities as thin as the electron gyroradius. Instead, it requires a spatial sampling area of the velocity distribution function of the order of few electron gyroradii. Indeed, we used here spatial bins having the size of $3\times6r_{Le}$ in the perpendicular plane to the magnetic field, comparable, for instance, with the

distance covered by MMS during 30 ms of flight within a plasma flow of 100 km/s at the magnetopause. If the VDF's sampling area is much larger than few electron gyroradii, the crescents will not be observed for electrons since the remote sensing effect will vanish at such large scales. We consider that the electron crescents produced by the mechanism discussed in this paper could be measured in-situ by the very high time resolution spectrometer on-board MMS in the vicinity of the magnetopause and in the presence of a deflected plasma jet.

The thickness of Earth's magnetopause is of the order of few hundreds of kilometers (or, equivalently, few proton gyroradii), as has been determined, for instance, from Cluster data (*Haaland et al.*, 2004). Thus, the magnetopause (which under certain circumstances can be approximated with a tangential discontinuity) is much larger than the electron gyroradii. Nevertheless, since the conditions for the formation of crescent-shaped electron VDFs do not require a TD as thin as the electron gyroradius, we believe that such distributions can be formed and measured at the Earth's magnetopause. To render more clearly the aspect of TD's thickness into our article, an explanatory text has been added in Section 2 (pg. 2, lines 38-41).

It should mentioned that for the particular set of parameters used in our simulations, the thickness of the simulated TD is comparable with few thermal electron gyroradii. Indeed, due to computational constrains, we considered $m_i/m_e$=36 and $V_{Te}/V_{Ti}$=30. Nevertheless, the kinetic effects discussed here should be observed also in the case of a more realistic input. This issue will be addressed in our future simulations.

**4.** *Related to the previous question: How about high-beta plasma effects? In this parameter regime, the electron gyroradii would be larger and easier to measure. Would there be any significant different for the proposed mechanism in that parameter regime? Have the authors tried such simulations? If not, why not? In such cases electrostatic effects such as polarization electric fields should be diminished. I think a brief discussion about this issue will benefit the readers by making the results of this manuscript more general and with a wider applicability.*

**Answer:**
The laboratory experiments of *Wessel et al.* (1988) have shown that both low and high-beta plasmoids do propagate across transverse magnetic fields through the effect of self-polarization, for $0.01 < \beta < 300$. Thus, the electrostatic polarization of *Schmidt* (1960) sustains the cross-propagation even for very high-beta plasmoids/clouds/jets. Nevertheless, to our knowledge, there are no detailed kinetic theoretical studies or full-particle numerical simulations on the topic of high-beta plasma jets interaction with non-uniform magnetic fields. Alongside with a comprehensive theoretical investigation of low-beta plasma jets, *Lemaire* (1985a, 1985b) discusses briefly the dynamics of high-beta plasma jets in the framework of the impulsive penetration mechanism. In this case, the diamagnetic effects are important and the geomagnetic field is strongly perturbed by the internal magnetic field produced by the plasma jet itself. As in the case of low-beta jets, the self-polarization electric field also sustains the cross-propagation of the high-beta jet.

We haven't investigated yet the problem of high-beta plasma jets, but it is definitely an important topic for us and shall be addressed in our future simulations. In these cases of high-beta plasma jets, the self-consistent magnetic field produced by the plasma jet itself is comparable with the background (external) magnetic field. This additional complication will

certainly modify the dynamics of the simulated particles. As a result, the velocity filtering effect could be directly affected, since the magnetic field gradient can change in terms of both intensity and scale. Moreover, we also expect that the remote sensing effect to be affected by the high-beta plasma regime. At this moment, we cannot estimate to what extend the magnetic perturbation produced by the high-beta jet will affect the formation of crescent-shaped electron VDFs. Supplemental simulations are required in this case. To render more clearly the aspect of high-beta plasma jets in our manuscript, an explanatory text has been added in Section 3 (pg. 6, lines 10-13).

**Answer:**
The main differences between the crescent-shaped electron VDFs obtained in our simulations and those reported in connection with magnetic reconnection sites at the dayside magnetopause are related to (i) the different physical mechanisms leading to their formation and also with (ii) the different spatial geometry involved in those processes.

The physical mechanisms proposed by, e.g., *Bessho et al.* (2016), *Shay et al.* (2016) or *Lapenta et al.* (2017), for formation of crescent-shaped electron VDFs involve meandering electron orbits in the vicinity of X-lines, thus in spatial regions characterized by a magnetic field reversal. In a similar geometry, *Egedal et al.* (2016) proposed that the perpendicular crescent-shaped distributions are produced by the diamagnetic drift of well-magnetized electrons in a thin boundary layer localized along the magnetopause separatrix. In our case, the physical mechanism we propose does not involve the spatial reversal of the magnetic field, since we consider a parallel tangential discontinuity. The crescent-shaped electron VDFs obtained in our simulations are formed by two combined effects: (i) the velocity filtering effect produced by the grad-B drift and (ii) the remote sensing effect of large Larmor radius electrons.

The electron crescents reported here are obtained in connection to a jetting plasma structure that moves in opposite directions tangentially to the parallel discontinuity surface

(our simulated magnetopause) and perpendicular to the magnetic field; they are found inside and also in the vicinity of the discontinuity towards the $-\vec{B} \times \nabla B$ direction (which is the $-\vec{M}$ direction in the (L,M,N) boundary-normal coordinates that are used, for instance, in *Burch et al.* (2016)). On the other hand, the electron crescents reported by *Burch et al.* (2016) have been observed in connection to the anti-parallel jetting along the $\vec{L}$ direction produced by magnetic reconnection at the dayside magnetopause; they are found close to the magnetic null region and along the $\vec{N}$ direction, as obtained also by, e.g., *Bessho et al.* (2016) and *Lapenta et al.* (2017). In the study of *Egedal et al.* (2016), the electron crescents are obtained in a boundary layer having the width of $\sim 1 r_{Le}$ all along the low-density separatrix. On the contrary, in our simulations, the electron crescents are obtained over a region in the perpendicular plane to the magnetic field that covers $\sim 12 r_{Le}$ along *Ox* (the $\vec{N}$ direction) and $\sim 20\text{-}30 r_{Le}$ along *Oy* (the $\vec{M}$ direction) (see Figures A2 and A3 in Appendix A of the manuscript). Also, the low-energy cut-off in the perpendicular velocity space reported by *Egedal et al.* (2016) is related to an exclusion perpendicular energy that varies with distance from the separatrix along the $\vec{N}$ direction, while in our simulations the low-energy cut-off varies with distance from the jet's centre along the $\vec{M}$ direction.

As emphasized in the concluding remarks of our manuscript: *"the crescent is a persistent feature of the VDFs that can be detected at relatively large distances from the interaction with the discontinuity and therefore might be used as a kinetic signature of such interaction"*. Thus, the crescent-shaped electron VDFs can be used to identify plasma jets that have interacted directly with a magnetic discontinuity, but additional information is needed as, for instance, data about the magnetic field, particle density or bulk velocity. A simple inspection of only crescents cannot discriminate between the different physical mechanisms leading to their formation.

To render more clearly this aspect into our manuscript, an explanatory text has been added in Section 4 (pg. 7, lines 36-40 & pg. 8, lines 1-8).

**3.** *I think that the authors should discuss the relevance of their results a little bit more. They mention that the distribution functions can be used to estimate the geometry of jets, but there should be further points to discuss. For example, I assume that the resulting distribution functions are unstable to wave generation. What was could be expected to be excited? And how would moment calculations, which assume Maxwellian distribution, be affected?*

**Answer:**

The velocity distribution functions obtained in our simulations are, indeed, unstable to wave generation. The ring-shaped electron VDFs produced by the velocity-filtering effect can generate, for instance, cyclotron harmonic waves (e.g. *Tataronis and Crawford*, 1970; *Lee and Birdsall*, 1978). It should be mentioned that the study of waves generation and instabilities in this simulation is beyond the scope of the present paper and shall be addressed in our future studies. Thus, it might be premature at the present time to approach this aspect into the manuscript.

Under certain circumstances, the moment calculation can be affected by the shape of the velocity distribution function. It has been shown by *Lee et al.* (2004) that the moment computation for non-gyrotropic (or crescent) VDFs may provide unrealistically large bulk velocities. Indeed, in such cases of non-Maxwellian distributions, large velocity moments can

be obtained even for a stationary plasma, if those distributions are produced by a remote sensing effect, as the one discussed also in this paper. Therefore, when calculating the moments of this kind of velocity distribution functions, both the distributions and moments should be examined carefully to establish if the large velocity moments really indicate an actual plasma flow or not.

**4.** *No ion distributions are shown in this paper. The authors could at least comment on if any effects of the interaction are shown in those.*

**Answer:**
The crescent-shaped VDFs are observed only for electrons in the current simulations, while the ion VDFs show no crescent features. Indeed, for the particular set of parameters used here, the ion grad-B drift velocity is smaller than for electrons and the corresponding velocity filtering effect is not large enough to produce an efficient scattering of ions along the $+\vec{B} \times \nabla B$ direction leading to an ion energy-dispersed structure with crescent-shaped ion distributions. Additional simulations are envisaged in the future to study the formation of crescent-shaped ion VDFs simultaneously with the electron crescents. To render more clearly this aspect into our manuscript, an explanatory text has been added in Section 3 (pg. 6, lines 14-15).

**Crescent-shaped electron velocity distribution functions formed at the edges of plasma jets interacting with a tangential discontinuity**

Gabriel Voitcu[1], Marius Echim[1,2]

[1]Institute of Space Science, Magurele, 077125, Romania
[2]Belgian Institute of Space Aeronomy, Brussels, B-1180, Belgium

*Correspondence to*: Gabriel Voitcu (gabi@spacescience.ro)

**BLUE** text = new text in response to **Referee #1** comments
**RED** text = new text in response to **Referee #2** comments

[revised manuscript text omitted]